# "I found out about Zika virus after she was born." Women's experiences of risk communication during the Zika virus epidemic in Brazil, Colombia, and Puerto Rico

María Consuelo Miranda Montoya[1], Claudia Hormiga Sánchez[2], Ester Paiva Souto[3], Edna Acosta Pérez[4], Gustavo Corrêa Matta[5], Marcela Daza[6], Gabriela Lopes Gama[7], Camila Pimentel[8], Marcela Mercado[9], Angélica María Amado Niño[10], Luz Marina Leegstra[11], Elena Marbán Castro[12], Olivia C. Manders[13], Lauren Maxwell[11]*

1 Facultad de Salud, Universidad Industrial de Santander, Fundación INFOVIDA, Bucaramanga, Colombia, 2 Facultad de Ciencias de la Salud, Universidad Autónoma de Bucaramanga, Fundación INFOVIDA, Bucaramanga, Colombia, 3 Interdisciplinary Centre for Public Health Emergencies (NIESP-CEE-FIOCRUZ), Oswaldo Cruz Foundation, Rio de Janeiro, Brazil, 4 Graduate School of Public Health / Hispanic Alliance for Clinical and Translational Research, Medical Science Campus, University of Puerto Rico, San Juan, Puerto Rico, 5 Interdisciplinary Centre for Public Health Emergencies (NIESP-CEE-FIOCRUZ), Center for Data and Knowledge Integration for Health (CIDACS/FIOCRUZ-Bahia), Rio de Janeiro, Brazil, 6 Grupo de Salud Materna y Perinatal, Instituto Nacional de Salud, Bogotá, Colombia, 7 Professor Joaquim Amorim Neto Research Institute (IPESQ), UNIFACISA University Center, Paraíba, Brazil, 8 Laboratory for Studies on Drugs, Vulnerabilities and Social Markers (LED), Department of Public Health (NESC), Aggeu Magalhães Institute—Fiocruz Pernambuco, Recife, Pernambuco, Brazil, 9 Dirección de Investigación en Salud Pública, Instituto Nacional de Salud, Bogotá, Colombia, 10 Facultad de Salud, Departamento de Salud Pública, Universidad Industrial de Santander, Bucaramanga, Colombia, 11 Heidelberger Institut für Global Health, Universitätsklinikum Heidelberg, Heidelberg, Germany, 12 ISGlobal, Hospital Clínic-Universitat de Barcelona, Barcelona, Spain, 13 Rollins School of Public Health, Hubert Department of Global Health, Emory University, Atlanta, Georgia, United States of America

* lauren.maxwell@uni-heidelberg.de

**Data Availability Statement:** Due to the sensitive nature of the information collected during the interviews, the institutional review boards (IRBs)

## Abstract

Providing accurate, evidence-based information to women with Zika infection during pregnancy was problematic because of the high degree of uncertainty in the diagnosis of the infection and the associated risk. The 2015–17 Zika virus epidemic overwhelmingly affected women in countries with limited access to safe abortion. Understanding women's perspectives on risk communication during pregnancy in the context of an emerging pathogen can help inform risk communication in response to future outbreaks that affect fetal or child development. We conducted a cross-sectional qualitative interview study with 73 women from 7 locations in Brazil, Colombia, and Puerto Rico to understand women's experiences of Zika virus (ZIKV) test and outcome-related communication during the ZIKV pandemic. We used thematic analysis to analyze the in-depth interviews. Participants in Brazil and Colombia reported that the healthcare system's lack of preparation and organization in communicating ZIKV test results and associated adverse outcomes led to their feeling abandoned and alone in confronting the challenges of a ZIKV-affected pregnancy. In contrast, participants in Puerto Rico reported that the regular testing schedules and clear, well-planned communication between the care team and between providers and pregnant

from each country where the research was conducted did not permit the researchers to upload de-identified transcripts of the interviews to a qualitative data repository. IRBs asked the researchers to take the following steps to ensure participants' confidentiality. These actions were presented to research participants in the informed consent. 1. The project researchers will not use this information for purposes other than those foreseen by the project. 2. Any data that could identify you will be excluded when disseminating the research results. 3. The collected material will be stored in protected digital files for at least five years, per (Resolution 466/12 and the guidelines of the CEP/ENSP in Brazil, Resolución 0830 in Colombia and 45 CFR 46.HHS regulation in Puerto Rico), access to which will be restricted only to the researchers involved in the research. Relevant, de-identified excerpts of transcripts are included in the manuscript text and supplementary files. Researchers external to the project team who want to review the original de-identified transcripts can solicit permission through the corresponding site IRB with the site PI in copy. Brazil: National Research Ethics Committee (CONEP) of Brazil, Biomedical Research Ethics: COMISSÃO NACIONAL DE ÉTICA EM PESQUISA – CONEP Phone: +55(61) 3315-5877 | conep@saude.gov.br Address: SRTV 701, Via W 5 Norte, lote D - Edifício PO 700, 3° andar – Asa Norte CEP: 70719-040, Brasília-DF PI: Gustavo Correa Matta Email: gustavo.matta@fiocruz.br Colombia Comité de Ética en Investigación Biomédica (CEIB) del Centro de Atención y Diagnóstico de Enfermedades Infecciosas Address: Calle 56 #36-74 Bucaramanga, Santander, Colombia Email: comitedeetica@cdi.net.co Chair person: Dr María Azucena Niño Tovar Comité Técnico Científico del Centro de Atención y Diagnóstico de Enfermedades Infecciosas Address: Carrera 37 #51-126 y calle 52 #36-19. Bucaramanga, Santander, Colombia Email: coordinadorainvestigacion@cdi.net.co PI: María Consuelo Miranda Montoya Email: mcmirandamontoya@gmail.com Puerto Rico Human Research Subjects Protection Office (HRSPO) School of Health Professions 2nd Floor, Suite 210 San Juan, PR 00936 E-mail: opphi. rcm@upr.edu Vanessa Sepulveda, MD, FACP IRB Director Phone: (787) 758-2525 ext. 2510 PI: Edna Acosta email: edna.acosta2@upr.edu.

**Funding:** This work was supported by a DFID/Wellcome Trust grant to the UNDP/UNFPA/UNICEF/WHO/World Bank Special Programme of Research, Development and Research Training in Human Reproduction (HRP), Department of Sexual and Reproductive Health and Research [grant number 216002/Z/19/Z]. This research also

women helped them to feel they could prepare for a ZIKV-affected pregnancy. Communication of the risk associated with an emerging pathogen suspected to affect pregnancy and developmental outcomes is a fraught issue. Public health authorities and healthcare providers should work together in the interpandemic period to understand families' preferences for risk communication during pregnancy in the presence of uncertainty and develop a community-informed plan for risk communication.

## Introduction

Risk communication is an essential component of health care, an important part of the health system's response to an emerging pathogen, and one of the pillars of outbreak response [1]. Healthcare communication includes both cognitive (informational) and emotional (acknowledgement of needs and feelings) components and must be contextualised to specific settings and individual and collective needs [2]. Healthcare communication should be clear, accurate, and ideally evidence- and values-based. Health systems and individual providers need to consider probability, level, and the exposure's effect on the individual, fetal, child, and community health [2].

While Zika virus (ZIKV) infection is generally subclinical, infection during pregnancy can have devastating consequences for fetal and child development. Despite global efforts to improve ZIKV diagnostics and to better understand the risk of adverse fetal, infant, and child outcomes, myriad uncertainties persist. Close to 80% of infections are asymptomatic, which reduces the accuracy of diagnostic assays, given that their accuracy decreases with the length of time from exposure [3]. WHO guidance on communication developed early in the ZIKV epidemic highlighted the need to prioritise access to comprehensive family planning services [4].

In this multi-country qualitative study, we explored how women who were pregnant during the 2015–17 epidemic were informed about diagnostic tests for Zika, the risks associated with the infection, and the diagnosis of Zika-related illness in their unborn child. In another manuscript, we summarise findings related to how women who were pregnant during and after the 2015–17 ZIKV pandemic preferred to receive diagnostic- and outcome-related information about the diagnostic test result and related outcomes in the presence of high levels of uncertainty.

We conducted this study as part of the ZIKV Individual Participant Data (IPD) Consortium individual participant data meta-analysis (IPD-MA) of ZIKV-related cohorts of pregnant women and their infants and children, initiated in 2017 [5]. Developing risk prediction models to better inform pregnant women and couples planning a pregnancy is one of the four key objectives of the ZIKV IPD-MA [6]. Results from this study are meant to support the development of community-informed models for risk communication to contextualize findings from the predictive models derived from the ZIKV IPD Consortium collaboration.

## Methods

### Design and population

Study sites were selected due to the high incidence of ZIKV infection during the outbreak, the burden of ZIKV-related congenital Zika syndrome (CZS), differences in access to contraception and legal abortion, and different levels of gender equity, economic resources, and religiosity. The study team included investigators at Carlos Albizu University in Puerto Rico, Oswaldo Cruz Foundation (FIOCRUZ) and the Instituto de Pesquisa Professor Joaquim Amorim Neto

received support from the Instituto Nacional de Salud, INFOVIDA and Centro de Atención y Diagnóstico de Enfermedades Infecciosas in Colombia. In Puerto Rico, this research received additional support from the Hispanic Alliance for Clinical and Translational Research supported by the National Institute of General Medical Sciences (NIGMS) National Institutes of Health (grant number U54GM133807). In Germany, this research received additional support from the ReCoDID Project, which is funded by the EU Horizon 2020 Research and Innovation Programme (grant agreement 825746) and the CIHR Institute of Genetics (grant agreement 01886-000) grants to LM. The funders had no role in study design, data collection and analysis, decision to publish, or preparation of the manuscript.

**Competing interests:** The authors have declared that no competing interests exist.

(IPESQ) in Brazil, Universidad Industrial de Santander, Instituto Nacional de Salud, Fundación INFOVIDA, and Universidad Autónoma de Bucaramanga in Colombia. Interviews were conducted in Campina Grande, Recife, and Rio de Janeiro, Brazil; Bucaramanga, Neiva, and Barranquilla, Colombia; and throughout Puerto Rico from Abril 2020 to July 2021 during the COVID-19 pandemic. Site-specific recruitment dates, source populations, and procedures for participant recruitment, interviews, and psychological support during and following the interview are reported in S1 Table.

Participants were women aged 18 years and older, living in *Aedes aegypti*-endemic areas affected by ZIKV during the 2015–17 outbreak, who were pregnant during the outbreak and who had and had not had ZIKV-affected children. Participants were recruited through health clinics, government services, community-based organizations, and social media organizations in Puerto Rico, through ZIKV cohorts in Colombia and Campina Grande, Brazil, and mother's organizations in Rio de Janeiro and Recife. In Puerto Rico, participants were recruited between 9 April 2020 and 1 November 2021. In Brazil, they were recruited between 13 November 2020 and 7 July 2021 and in Colombia between 9 February 2021 and 24 June 2021. The study was conducted on a rolling basis to allow for the iterative evaluation of data and ensure that emerging themes could be identified and explored.

## Data collection

The study team collaboratively developed separate semi-structured interview guides for women who did and did not experience an adverse ZIKV-related pregnancy outcome. Guides were developed by reviewing the ZIKV and risk communication literature and discussing ZIKV infection during pregnancy with the study team and our colleagues in the social sciences and ZIKV-focused research. The guides were divided into five sections: current experiences with the COVID-19 pandemic, knowledge about Zika, perceptions of risk, actual and ideal delivery of Zika results to women pregnant during Zika, and use of de-identified patient data. The guides were pilot-tested in Brazil among women who would have been eligible for inclusion in the study. Modifications, including shortening the guide and clarifying the section on data sharing-related attitudes, were incorporated into the documents. Interview guides are available on the Open Science Foundation (10.17605/OSF.IO/BZE8C).

**Data collection.** In-depth interviews (IDIs) were conducted via Zoom or WhatsApp, depending on participant preference. The one-time interview, including the five study objectives, lasted one to one and a half hours. Interviews were conducted until saturation was reached within strata (geographical location, women pregnant during ZIKV with and without affected children), i.e., no new themes emerged [7, 8].

**Analysis.** IDIs were conducted in Portuguese and Spanish and transcribed and analyzed in the original language. Deductive codes were developed from the interview guides; the study team memoed and discussed each transcript to develop inductive codes based on those conversations. Transcripts were coded thematically using Dedoose software [9], and we held weekly small group and team discussions to ensure consistent interpretation of the codes across the study sites and teams. We provide the original and English translation versions of all quotes and additional supportive quotes in S2 Table. The codebook is included in S3 Table.

**Researcher characteristics and reflexivity.** The study team included 17 researchers from various professional backgrounds, including community-based participatory research, sociology, counselling, qualitative and mixed methods research, ID diagnostics, and epidemiology. All researchers are fluent in Spanish or Portuguese; transcripts were discussed and analyzed in their original language. Researchers were from included countries except four team members and had worked closely with study participant source communities.

### Research ethics

The study protocol and forms were reviewed and approved by the Ethics Review Committee (ERC) or Institutional Review Board (IRB) of each site involved in this project before initiating data collection. These included the WHO ERC, the Emory University IRB, National Research Ethics Committee (CONEP) of Brazil, Biomedical Research Ethics Committee of Centro de Atención y Diagnóstico de Enfermedades Infecciosas-(CEIB-CDI), the Ethics Committee and Research Methodology of the Colombian National Health Institute (CEMIN-INS) of Colombia, and the BRANY SBER IRB of Puerto Rico, contracted by the Universidad Carlos Albizu. For remote interviews (29 from Brazil, 18 from Colombia, and 24 from Puerto Rico), participants completed verbal informed consent; for in-person interviews, participants provided written informed consent (2 from Brazil). Further ethical concerns are reported in the PLOS One Inclusivity in global research questionnaire (S1 Text).

## Results

We report the demographic characteristics of the 73 women from 7 sites who participated in the study (31 from Brazil, 24 from Puerto Rico, and 18 from Colombia) in Table 1. More than half of the participants (55%) had a child with at least one of the signs of CZS, while 45% had a child who had not been diagnosed with CZS. Regarding education levels, the majority across all regions had a graduate level of education, with Brazil at 42%, Colombia at 39%, and Puerto Rico at 54%. High school education was prominent in Brazil (38%) and Colombia (22%), while in Puerto Rico, 17% had a high school education. Most women were between the ages of 29 and 34.

### Diagnostic testing for Zika infection and communication of results

Symptomatic participants in Brazil and Colombia generally reported receiving a clinical diagnosis of ZIKV rather than a laboratory test. In several cases, women were not tested for ZIKV or were not told they had had ZIKV until after an abnormal ultrasound late in pregnancy or after they had given birth to a ZIKV-affected infant. Pregnant women in Brazil and Colombia who reported their symptoms or who were tested for ZIKV and received a positive result were told "not to worry," that ZIKV was "like an allergy" or "similar to Dengue" and was not like

**Table 1. Research participants' demographic characteristics (N = 73).**

|  | Brazil (N = 31) | | Colombia (N = 18) | | Puerto Rico (N = 24) | | Total (N = 73) |
|---|---|---|---|---|---|---|---|
| **Pregnant during ZIKV** | N | (%) | N | (%) | N | (%) | |
| Mothers of children with CZS | 21 | (68) | 11 | (61) | 8 | (33) | 40 |
| Mothers of children without CZS | 10 | (32) | 7 | (39) | 16 | (67) | 33 |
| **Average age** | | 34 | | 29 | | (30) | |
| **Level of education** | | | | | | | |
| Higher than college | 13 | (42) | 7 | (39) | 13 | (54) | 33 |
| Any college | 0 | (0) | 7 | (39) | 4 | (29) | 10 |
| Any high school | 12 | (38) | 4 | (22) | 5 | (17) | 21 |
| Any primary or middle school | 6 | (19) | 0 | (0) | 2 | (8) | 8 |
| **Marital status** | | | | | | | |
| Married | 15 | (48) | 5 | (28) | 9 | (38) | 29 |
| Cohabitating | 13 | (42) | 7 | (39) | 9 | (38) | 29 |
| Single | 3 | (10) | 5 | (28) | 6 | (25) | 14 |
| Divorced | 0 | (0) | 1 | (6) | 0 | (0) | 1 |

rubella or other serious exposures during pregnancy. In some cases, symptomatic women who asked to be tested said they never received a test (Quotes 1–2, S2 Table).

*At 2 months, I got Zika. I had an outbreak, and I went to the emergency room to get tested, but even I went to the emergency room, and they told me to take acetaminophen, and that's it, just go home, that is, they told me no. I requested the test, and they told me no, so I told him that I was scared because I was pregnant and wanted to do the test and they told me no, that there was no problem, that I should take acetaminophen and nothing else, but yes, to one gives you the uncertainty, but well I was already pregnant and everything and well the truth at the time it hit me, I took it as a passenger, it passed me by and I didn't think about it again, that was almost seven months was when he began, let's say, to suspect something and all that. Before, it was a completely normal pregnancy.*

(CZSN004, Colombia, Mother of a child with CZS, under 25)

*When I got pregnant with [child's name], I was 12 weeks, and I started. . . I developed a really bad itch, and this itch got so. . . bothered me so much that I had to go to the ER and then to the ER. . .and I asked him how this could affect my baby. The doctor didn't say anything, that it was like dengue and that it was going away. He gave me an allergy medicine and some ultrasounds. I took the allergy medicine. I had this itch for four days. I had the ultrasounds, and everything was normal.*

(BRID_SPECP12, Brazil, Mother of a child with CZS, age 25–34)

In contrast, in Puerto Rico, where the ZIKV epidemic began later than in Brazil and Colombia and which had the full financial and laboratory support of the US-Centers for Disease Control and Prevention (CDC), ZIKV diagnostics were incorporated into routine prenatal care [10]. Health providers asked that women bring someone to accompany them when receiving the test result. Women who received a positive test result were immediately referred to a specialized care center for weekly follow-up visits. In contrast to Brazil and Colombia, women in Puerto Rico reported that the possible consequences of infection in the child were explained to them in detail (Quotes 3–4, S2 Table).

*The first tests. . .well, everything went well. One day, I had a routine appointment with the gynecologist, and he. . . I told him that he had a rash all over his body. So, the doctor prescribed a soap for my skin to calm the itching and what the rash was and gave me some antibiotics. Then he told me: "We are going to repeat the Zika test because you are going to be entering what would be the third trimester to repeat it before [you give birth]." When the results come in, they tell me: "Mom, you came out positive for Zika."*

(ID304, Puerto Rico, Mother of a child with CZS, age 25–34)

*He told me, "Here, in all these, you came out negative. In the only one you came out positive was Zika, so you have Zika". From there, I opened my eyes very wide, despite the fact that I already suspected it, because I had a little faith that it wasn't, that it was just silly what I had had and it wasn't that. So, there she told me the [public hospital] protocol. So now I had to go to a perinatologist who specializes in pregnant moms with Zika.*

(ID305, Puerto Rico, Mother of a child with CZS, age 25–34)

**Explanation of ZIKV test results and associated uncertainty.** Some participants in Brazil and Colombia received a ZIKV test but were never given the results. In Brazil and Colombia, where results were often delivered by health authorities rather than the women's health care provider, when they did receive the results, women reported that they received minimal information on what the results could mean and no information related to uncertainty in the results or the associated risk (Quotes 5–8, S2 Table).

*I think that they also have to explain to one, surely one does not know what effects it brings, because to me, for example, they only told me that I had Zika and that it was a disease that could affect the baby, that it deformed it, that was all.*

(NOCZS008, Colombia, Pregnant during ZIKV, not adverse pregnancy outcome, 35 or over)

In other cases, participants who were tested were simply informed that their result was "positive" or that "everything was fine" but did not receive additional information or have access to the test report. While some participants in Puerto Rico also reported receiving limited information about the meaning of the test result, most participants in Puerto Rico said that they reviewed and discussed the results with the healthcare provider.

In Colombia, several participants whose children were born with CZS were not told about the results of the ZIKV laboratory test, which led to uncertainty about the cause of their infant's neurological disorder.

*They (health personnel) said [the microcephaly] was due to Zika, they immediately said that they associated it with Zika, but in itself, there is no blood test that certifies that it is due to Zika, that is, since he was born in the month of Zika children, they put him there with them, but there is no blood test that tells me if it is due to Zika.*

(CZSBQ002, Colombia, Mother of a child with CZS, under 25)

In Brazil, several mothers did not receive information about their ZIKV status until after giving birth to a ZIKV-affected child. At that point, the diagnosis was irrelevant to them (Quotes 7 and 8, S2 Table).

*So, I already knew the result because after he was born, we saw that he had microcephaly. I already knew it was from Zika, the result was the least important, the positive result for Zika was less important, as I already knew, I knew that I had microcephaly.*

(SPECP01, Brazil, Mother of a child with CZS, age 25–34)

*When this result came out,rI already knew he was already a year and a bit old; he was one year and four months old, more or less [child's name], and we were already in treatment. . . . she had the tomography done and came to the doctor, so she (the doctor) said that [child's name] had cerebral palsy and microcephaly and had calcifications; well, then, she already knew that [child's name] had microcephaly, then after in 1 year I found out that it really was, it was because of Zika.*

(SCGBS06, Brazil, Mother of a child with CZS, under 25)

In several cases, women gave birth to an affected child after being told they had tested negative for ZIKV. Almost no participants reported being informed that they could have ZIKV, although the test was negative or vice versa. One participant from Brazil who had received

multiple ZIKV tests said that they felt that the conflicting test results reflected the moment's uncertainty.

When women were diagnosed with ZIKV infection towards the end of pregnancy, they were generally told that their infant had passed the relevant developmental stages and they should not worry. (Quotes 9–10, S2 Table).

### Diagnosis of Zika-related neurological disorder

The diagnosis of fetal neurologic impairment attributable to Zika infection was based on ultrasound findings, specifically microcephaly and cerebral calcifications, whether or not there was a clinical or laboratory history of maternal, fetal, or infant Zika infection. In Brazil and Colombia, ZIKV-related neurological complications were always identified late in pregnancy, regardless of whether women were receiving special follow-up related to their ZIKV diagnosis.

For women who presented with ZIKV-type symptoms during pregnancy, the fetal neurological complications became the de facto confirmation of maternal Zika infection. Many women were asymptomatic, and, in some cases, both asymptomatic and symptomatic women who had affected children had received a negative test result. In these cases, the neurologic abnormalities identified through a late-stage ultrasound were a complete surprise. A few women in Brazil and Colombia were pregnant early on in the pandemic and did not learn about ZIKV until after giving birth to an affected child (Quotes 12–15, S2 Table).

*When I got pregnant that, I was infected with this virus, they only told me it was Chikungunya, that they didn't know what could happen. I reported that I was pregnant, and then nobody could tell me what could happen and then I had a little bleeding, and I went ahead with the pregnancy; it came out in the twelfth week of pregnancy, so since nobody knew anything about it, there was no way to knowing what the pregnancy, fetus, postpartum, or whatever would be like, everything was very surprising for me, for me I only found out about the Zika virus after she was born, which was when they also began to be born in the Northeast, all with microcephaly, and when they began to do the tests, they saw that it was actually microcephaly due to the Zika virus.*

(SRJEP03, Brazil, Mother of a child with CZS, 35 or over)

Women in Brazil and Colombia who were diagnosed after ZIKV was known to cause fetal abnormalities but early in the epidemic were confronted with unprepared providers who were themselves surprised by the ultrasound findings.

Then he (gynecologist) calculated [the head circumference] and calculated it. "It just doesn't work, it doesn't work, the head is not the size it should be for the time of pregnancy." When he looked at the ultrasound, nothing, nothing. He called another primatologist who was there, "Maybe I was wrong, maybe I did something wrong, let's confirm." And he came back, came back, and the other man also looked and nothing, definitely not. Then another one came and also says the same. So, they stayed, just like they hadn't had, just like, like many cases like that. So close, they began to worry, then to explain to me what had happened. "This is what happened." He asked me, "Did you have a fever? Did you have any symptoms of that?" He asks me the questions and I did. When I was eight weeks pregnant I got a fever, spots like that, pain in the joints, a headache, but they never told me anything. I went, and they never told me anything that I had to look at or do a deeper follow-up, that is, nothing different from an ultrasound. . . So, the doctor immediately told me no, that the baby most likely came with microcephaly because, how is it? The little head was not, her

brain it was not fully developed, it was smaller than the time, that for the time of pregnancy that I had, it did not have the size that it was. And so, he sent me a series of tests to rule out. . .. It was the first time they sent me to do a blood test to find out if I had had Zika or not, that same day they did it.

(CZSBQ001, Colombia, Mother of a child with CZS, age 25–34)

*Doing the ultrasound on him (gynecologist), some, some, uh. . . some measurements did not add up, he told me, "No, it's that suddenly my ultrasound machine was damaged," he came back and did it, came back, erased, turned off and when he told me, "we have to do some more in-depth tests, suddenly my ultrasound machine is failing me and well I have to do something to him", then I told him how come? Then he told me that some measurements of my son's head circumference did not add up, eh in his little head, "he explains more in Spanish" I told him and that well, we had to carry out some tests, then he gives me the formula, I he gave the orders for me to have them done by the EPS and, well, that was in a matter of 15 days, because of the priority, he did give it priority and after 15 days they did it for me. . .the doctor, thinking that I already knew the diagnosis, told me, "yes, indeed the child has his calcifications, the child has microcephaly and tatata. . .tatata" and he tells me everything as one, right?*

(CZSBUCA002, Colombia, Mother of a child with CZS, age 25–34)

In one case, a participant in Brazil with an abnormal fetal ultrasound later in pregnancy felt that her provider did not give her information on microcephaly and she had to depend on the media for information.

*We were already following it [the outbreak], with fear, but I had great faith that nothing was going to happen. I was still in the ultrasound room, right, when the difference came, she said, really now there was a difference in head circumference. And there we already knew what microcephaly was, but she didn't explain much either. It was her first case. . . here in Rio, it was the first case in this outbreak that she was caring for a woman, a child, no? with microcephaly. So, she also didn't explain much to me; it was only on the media.*

(BRID_SRJEP06, Brazil, Mother of a child with CZS, age 25–34)

Most Colombian and Brazilian interviewees whose children had adverse ZIKV-related neurological outcomes were told about their child's condition towards the end of pregnancy or at birth. The ZIKV test was ordered following the detection of an abnormality on ultrasound rather than vice versa. Colombian and Brazilian mothers reported that not being told earlier prevented them from preparing emotionally and logistically for their child's disability. In Colombia and Brazil, women reported a lack of continuity in health care providers wherein the provider who interpreted the ultrasound was not the provider engaged in ZIKV-related follow-up. The lack of continuity caused gaps in communication and information that exacerbated the already difficult situation for mothers with affected pregnancies (Quotes 16–18, S2 Table).

*E: Did you hear of what could happen if a pregnant woman had Zika?*

P: When I used to talk about microcephaly because when I saw on television that I was most likely to have these children with microcephaly, it was something that I had never seen talk about microcephaly, never in my life, I was very afraid.

She was there, it was when she (doctor) said "she was a child with microcephaly", she was just doing it, so she didn't need to do anything because she was desperate at the time, right? I didn't know what microcephaly was, and she didn't explain anything to me, she explained it to me, "oh, it's like that", maybe I used oil at that time. . . I'm tempted to get rid of the magic, but I don't eat and eat very brave with that doctor, I wanted to place all the blame on her, you know, how she felt that she was destroyed or my dream.

(BRID_SCGBS01, Brazil, Mother of a child with CZS, 35 or over)

*E: Who were you with that day when they explained everything about your baby?*

P: With my husband. . . It turns out that the first day I went with my oldest girl to meet the baby, but since I didn't expect to be given that news, I got sick, right? Then I got very sick and it left when the doctor explained it to me, because I began to see her face, I saw that she was talking to the other doctor and that I saw that something was wrong with them, I began to suspect. When they explained the situation to me, I was not well, I got sick and left the consultation. I got out and went to the street, because I got really sick, so this. . . and I didn't want my older girl to see what was happening, I didn't want her to notice her, because I didn't know how to explain it to her. I felt bad because she had gone to meet the baby, and we couldn't show it to her.

(CZSBQ001, Colombia, Mother of a child with CZS, age 25–34)

*. . . they [doctors] always told me that this was a new disease, that it was a disease that was not known, that they were not certain that it was going to happen, that in all the children it was behaving differently, that it had to be prepared for the good and for the bad, that I not have so many illusions.*

(CZSBUCA002, Colombia, Mother of a child with CZS, age 25–34)

In contrast, interviewees who had tested positive during pregnancy in Puerto Rico, where regular ZIKV testing was incorporated into routine prenatal care, were immediately referred to specialized care, facilitating early access to ultrasound and allowing women to prepare better to welcome their ZIKV-affected infant (Quote 19, S2 Table).

*There was a boom, the outbreak was like there in Brazil and the media reported that the problem that the mothers were having when. . . was the problem of the babies, mainly that they were having problems when they were born. So, what she [doctor] told me was that she sent me to the perinatologist all the time. She did not give me a regular sonogram because the perinatologist sees everything more specifically if there is any defect or inefficiency in the baby. That's why the care was more intense compared to my other pregnancy, which was normal. It was very intense and she was very meticulous. She was very precise in everything.*

All interviewees with a child affected by Zika described the personal challenges they faced when receiving information about their child's condition. However, those who received information in private, in a calm and considerate manner, felt that the deliberate, thoughtful way the information was provided helped them cope with the difficult situation.

(ID305, Puerto Rico, Mother of a child with CZS, age 25–34)

**Interaction between delivery of information and women's cultural beliefs.** In delivering information about their fetus's condition, women highlighted the importance of providing the

information in a way that respects their cultural and religious beliefs, especially around abortion (Quote 20, S2 Table).

*He took me to his (doctor's) office, outside, where they did the ultrasound; we sat down, he talked to us, he talked to us, he told us the things, let's say positive, the negative things that the girl's condition could have, it was sincere from the beginning, because he didn't tell me, "no, the girl is bad, and you won't be able to do anything". He told us the girl is not doing well, but I can't tell you what God's will is either. When they talk to you like that, even though you're in pain, it's different from being told: "No, No, the girl is bad. . . no, look, you have to go somewhere and go and do this, I, I don't know, now it is your decision". But he, however, told us: you have the right to an interruption of the pregnancy if you want if you want, it is not an obligation. . . . as I say, [the doctor] was very subtle in saying things, so I say that to explain that to him so that one does not remain so, so traumatized.*

(CZSBQ001, Colombia, Mother of a child with CZS, age 25–34)

Women who were told that it would be better to abort the pregnancy because of the severity of the fetal condition felt that the way that the information was delivered was itself traumatizing and referred to the difficulty of being presented with the fetal anomaly and suggestion to abort at the same time without time to digest the information related to the fetal condition (Quote 23, S2 Table).

*It was, "He has microcephaly, look", so I said, "Yes, and now what do I do?" then he said, "Look, I'll be honest, your son will not walk; he; he will not speak, he will not listen, he will live in a vegetative state on top of a bed, and I sincerely believe that he will not live more than one year. If I were you, I would ask God to take him away from me because you are very young, and he is going to interfere in your life; maybe you will have the opportunity to have other children." Then I arrived, I was like, looking at his face, because then I didn't need that diagnosis, he didn't have to say that. . . you understand? I just wanted to know what I would do from then on, look for the specialists. . . .*

*Then he just threw it away, put the paper on top of my gurney, turned around. . . when he turned his back to me, there was this scream, like, you know, from the inside, "Hey," so I said, "What professional quality are you? ". . .then I know that I began to argue with him; I told him "and this is not the procedure of a professional, of simply coming and telling a mother what she is going to be, how long she is going to live, I did not ask her for that, I asked him from now on what I should do and who I should look for."*

*. . ."Why don't you ask for other tests so that we can confirm them? Now, you get to throw a bomb at me; you turn around and walk away. That's not how things work, you see, friend. Learn to be a professional because you chose your profession, but my son and I did not choose to be in this situation, and you have to treat us as people". . .I ended up crying more because of the way he spoke to me than because of the diagnosis with me, you know?*

(BRID_SPECP12, Brazil, Mother of a child with CZS, age 25–34)

How women who learned that they had a ZIKV-affected fetus considered and felt about the option of aborting the pregnancy varied by the way the information was delivered, abortion access issues, and their own religious and personal beliefs. In Colombia, participants felt that providers presented the option to abort at the same time that they learned about the ZIKV-affected pregnancy, and they insisted that abortion was the "best" option. They felt that the information was delivered in an insensitive way without feeling, making it harder for them to decide.

When Colombian women did decide to terminate a ZIKV-affected pregnancy, they found that pregnancy termination was not available given their gestational age and barriers to abortion access. In contrast, women from Puerto Rico and Brazil felt that the option to terminate the pregnancy was presented in a calm rather than an abrupt manner, which allowed them to consider whether to terminate the pregnancy (S2 Table, supportive quotes 22–24). Participants highlighted religious beliefs and the difficulty of having a disabled child as major factors in considering pregnancy termination (S2 Table, supportive quotes 25–26).

## Discussion

Risk communication during pregnancy in the presence of uncertainty in both the accuracy of the diagnostic and the probability of an adverse outcome is a fraught issue and becomes especially complicated in countries with inequitable access to vector control measures, contraception, and safe abortion. Risk communication during pregnancy must incorporate local values and realities around access to safe, quality abortion. In this multicountry, cross-sectional qualitative study, we learned from 73 women who were pregnant during the 2015–2017 ZIKV epidemic about their experiences receiving, or not, their ZIKV test results and information about the expected or actual effects of ZIKV exposure on their pregnancy.

We found that while women's experiences of risk communication were relatively stable within countries, there were important cross-country differences in women's experiences. Women who were pregnant during ZIKV in Brazil and Colombia frequently were not tested for ZIKV or did not receive their ZIKV test results and learned that they had a ZIKV-affected pregnancy during the first abnormal ultrasound scan. In contrast, in Puerto Rico, where the epidemic wave was later than in Brazil and Colombia and which received significant logistic and financial support from the US-CDC, ZIKV testing was incorporated into routine clinical care, and results were regularly and systematically communicated to participants. Similarly, while participants in Brazil and Colombia reported being informed abruptly about their ZIKV-affected pregnancy and not connected to resources to either make the decision about pregnancy termination or determine pathways for supporting their ZIKV-affected child, those in Puerto Rico said that they were asked to bring their partner or another support person when learning about their ZIKV test result and women with ZIKV-affected pregnancies were immediately referred to specialized follow-up and related resources. While the health system response in Brazil and Colombia amplified women's distress upon learning about their ZIKV-affected pregnancy, women in Puerto Rico calmly received more comprehensive information. Participants in Puerto Rico were more likely to learn their test results and the associated risk from a coordinated healthcare team than women in Brazil and Colombia. This difference may be caused by the fact that the ZIKV epidemic began later in Puerto Rico than in Brazil and Colombia and because, in addition to time to prepare, Puerto Rico was able to leverage resources from the US-CDC.

## Strengths and limitations

The main strengths of this research are the transnational composition of the sample and the research team. The diversity of the sample made it possible to compare women's experiences across health systems and different contexts within and between countries. This study had some limitations. The gap between the Zika epidemic and fieldwork may have made it difficult to recall details of the ZIKV testing process or risk communication during pregnancy. However, this fact made it possible to capture the reinterpretation that the participants made of their experiences, in light of the lessons learned from raising ZIKV-affected children. Another limitation was the impossibility of conducting face-to-face interviews due to the COVID-19

pandemic. Other studies suggest that participants can meaningfully engage in remote qualitative interviews [11]. Lastly, the time between data collection and publication of our findings is a limitation. Like the participants, our team found ourselves on the front lines of COVID-19 response while still trying to complete this study.

## Differences in care across countries

Risk communication differed importantly across countries. Participants who were pregnant during the ZIKV epidemic in Brazil and Colombia generally reported a clinical diagnosis of Zika with no subsequent initiation of special follow-up and no communication about the risks associated with a positive result or the uncertainty associated with the diagnostic. Few said that they had been tested when they presented with symptoms. Some participants reported asking to be tested for ZIKV because they recognized the signs of infection from information received from mass media. Participants reported that they were not tested then, but that "more specialized" ultrasounds were performed later in pregnancy. In contrast, participants from Puerto Rico reported repeated laboratory tests during pregnancy and participants who received a positive test result were immediately referred to a specialized care center.

Participants did not report being informed of the possibility of having ZIKV infection despite having a negative result or vice versa. Some women who were pregnant during the epidemic and whose children did not present clinical evidence of neurological alteration at the time of the interview were informed about the possibility that their children would not be affected by the infection after receiving a positive test result because the infection occurred at a late stage of gestation or because they did not present symptoms. Participants were sometimes told that ZIKV infection during pregnancy was not an important concern, like an allergy, and was unlikely to be related to adverse pregnancy outcomes. While we found significant differences in women's experiences of learning about their test results and related fetal or infant outcomes, we did not find such differences across sites within countries.

## Country-level differences in ZIKV-related testing, follow-up and risk communication

The findings on the diagnostic process of Zika in pregnant women and of neurological alteration in their children mirror the differential care trajectories across the three countries during the 2015–17 ZIKV epidemic. In Colombia and Brazil, although some of the women interviewed were diagnosed with ZIKV via a laboratory test and referred for specialized follow-up, most participants reported difficulty in accessing diagnostic tests for ZIKV or were tested without being given the time result. Most women from Brazil and Colombia with ZIKV-affected pregnancies learned about fetal neurological alteration late in pregnancy. Their experience contrasted with Puerto Rico, where ZIKV testing was incorporated into routine care and the health system had time, and perhaps the funding and political will to implement a supportive, transparent system of communication and equitable connection to follow-up services for women with affected pregnancies.

These country-level differences can partly be explained by the spatiotemporal evolution of the ZIKV epidemic, which initially emerged in Northeastern Brazil and then spread to Colombia and later to Puerto Rico. As the epidemic progressed, other countries in Latin America and the Caribbean gained experience and ministries of health services developed more comprehensive training and educational resources for health personnel, patients, the general population, and social media campaigns. This evolution corresponds to the paradigm proposed by Débora Diniz of "generations" of women affected by ZIKV, the first generation being those who learned the virus was circulating by discovering it in their own bodies [12].

## Country-level responses to the 2015 ZIKV epidemic

As the first country to respond to the 2015 epidemic, the Brazilian Federal Government prioritized vector control actions. In addition, resources were invested in the care and reception of people affected by the ZIKV and microcephaly in the Unified Health System (Sistema Único de Saúde; SUS). Simultaneously, research initiatives were promoted with the participation of the public health surveillance system and national and international institutions, seeking to accelerate the development of innovations for the response to the virus and the development of communication strategies for different audiences [13]. Despite the government's efforts, structural actions, including basic sanitation and water supply issues, access to contraception or safe abortion care, and the gender, race and class dimensions of epidemic response were not well addressed.

Following actions in Brazil, Colombia implemented the Zika Fever Response Plan, which focused on strengthening surveillance systems for vector-borne diseases and increasing health education, prevention, and vector control. The plan includes guidance for health professionals on prevention strategies, diagnostic criteria, follow-up in suspected or confirmed cases, counselling to guide voluntary termination of pregnancy, and psychosocial support, among others [14, 15]. However, the plan was implemented in 2016, after the epidemic peak in Colombia. Non-standardized health care during much of the epidemic, as well as the country's testing protocol, which required samples be sent to the National Institute of Health in Bogota, the only reference laboratory for ZIKV diagnostic tests, meant that test results could not be returned in a timely fashion [15]. These barriers to testing and risk communication were reflected in women's accounts of their experiences with risk communication during ZIKV in Colombia and are similar to the findings from other research conducted with Colombian women who were pregnant during the ZIKV epidemic [16, 17].

Puerto Rico, as a US jurisdiction, received support from the US-CDC and was part of the Zika Virus Preparedness, Response, and Recovery Plan from April 2016. The plan included staff training, educational materials for pregnant women, community outreach strategies, guidance for travelers on and off the islands, and Puerto Rico developed a vector control and laboratory testing action plan. Additional strategies focused on preventing sexual transmission (media communications and community education) and surveillance and monitoring of pregnant women for follow-up of positive Zika tests, including for blood donations [18]. In a multisectoral effort, the CDC Foundation, the Department of Health, Puerto Rico Obstetrics & Gynecology, the Primary Health Association of PR, the University of Puerto Rico, and other community, private, and public organizations carried out the Zika Contraceptive Access Network to provide women in Puerto Rico with free long-term contraceptive alternatives in a single medical appointment. Analysis of this multisectoral initiative demonstrated that access to effective contraceptive methods and other reproductive health services, which were and continue to be very limited in Puerto Rico, reduced costs of care, increased access to a full range of contraceptives, training of health care providers on counseling, patient-centered contraceptive practice, and implementation of a robust public education campaign [19]. Evaluations of the plan's implementation, conducted with CDC leadership, demonstrated the importance of collaboration between CDC and US state and territorial government offices in strengthening capacity to respond to this emerging public health threat [18].

The temporal and spatial evolution of the epidemic meant that Puerto Rico had additional time to develop a comprehensive response, which was likely further supported by the large, well-funded infrastructure of US-CDC. Puerto Rico's approach of including ZIKV testing as part of antenatal care and early referral of women with affected pregnancies to specialized care, which continues today differs meaningfully from the situation in Colombia and Brazil [20].

## Gender inequities and Zika

As with ZIKV, ID outbreaks disproportionately affect the most marginalized groups who lack equitable access to preventative measures, diagnostics, and care [19]. Addressing complex emergencies requires a human rights approach with a gender perspective, which becomes even more critical for vulnerable populations, including pregnant women [21]. Analyses of the Ebola and Zika responses identified gaps in addressing structural gender inequities and an over focus on preventative measures (e.g., safe sex, bed nets) with the assumption that different groups of women are equally able to make these decisions [21]. Autonomy to make sexual and reproductive health decisions is framed by socially established gender norms and closely related to socioeconomic status meaning that those at highest risk of ZIKV infection in pregnancy were also those least likely to be able to implement preventative measures or to access safe abortion care in a restricted setting [15]. In Brazil, the ZIKV public health campaigns focused on women's need to prevent unwanted as a means to reduce ZIKV vulnerability rather than men and women's shared responsibility [22, 23]. In Colombia, the most developed component of the Zika Fever Response Plan was vector control. At the same time, sexual and reproductive health consequences were considered at the national level but not operationalized in local contexts. In addition, the Colombian Zika Fever Response Plan was not aligned with the National Policy on Sexual and Reproductive Rights or the Ten-Year Public Health Plan in force during the ZIKV epidemic [15].

The wealthiest women have both better access to birth control and safe abortions, which is particularly important in countries with restrictive abortion laws like Brazil and Colombia [21, 24]. However, Colombia recently decriminalized abortion up to 24 weeks of gestation [25], making Colombia more similar to Puerto Rico where abortion was permissible for the health of the mother, including mental health and socio-economic well-being during the ZIKV epidemic [26]. an early ZIKV diagnosis may facilitate access to safe abortion and is essential for early women who, for religious reasons, prefer not to terminate their pregnancy as they need time to prepare emotionally and to identify resources internal and external to their families and communities to support best their ZIKV affected infant. Women universally reported distress and feeling that their cultural and religious beliefs were disrespected when they were presented with the option to terminate the pregnancy alongside the abnormal ultrasound findings. That said, we did not directly explore the connection between attitudes towards abortion and risk communication in this study. Understanding and addressing religious and cultural concerns around contraception and abortion and how those concerns are encoded in laws and policies to ensure unstigmatized access to high quality contraceptive and abortion care is an important part of ensuring addressing emerging infectious diseases that affect pregnancy and fetal development. How abortion-related access and beliefs affect preferences of WRA and their partners for learning about the risks of emerging pathogens suspected to affect fetal or infant development before the next epidemic of an emerging infectious disease was not explored here and should be explored in future research.

Particularly in Colombia, termination of pregnancy was reported as a hastily communicated option offered immediately following the news of abnormal ultrasound, which women saw as insensitive and disrespectful of their personal and family beliefs regarding abortion. Women's preferences for discussing pregnancy termination in the context of risk communication need to be carefully considered in the cultural, religious, and legal context [15]. Our findings that many participants saw raising their ZIKV-affected infant as an expression of God's will are similar to qualitative research related to pregnancy termination decision-making in the context of the ZIKV epidemic conducted in Brazil, Puerto Rico, and mainland US and are especially important in the Latin American community where faith and spirituality are closely

linked to health care decision-making [27]. Women need to have enough information and time to make an informed decision about terminating a pregnancy, free from external pressures or restrictions. The lack of support for the mother's mental health, supporting the mother together with their ZIKV-affected child, and support for families with ZIKV-affected children were highlighted as significant issues by Brazilian and Colombian study participants. Discussing the emotional and physical consequences for mothers and families and the lack of adequate financial and therapeutic support for ZIKV-affected children were significant sources of distress for Brazilian and Colombian mothers as they described their experience learning about their ZIKV affected pregnancy. In the healthcare response to an emerging pathogen that affects fetal development, women should have the support of the state for both continuing their pregnancy and raising their affected child or terminating their pregnancy.

## Conclusion

Women who were pregnant during ZIKV in Puerto Rico tended to report more neutral or positive experiences, while women in Colombia and Brazil felt negatively about how the ZIKV diagnosis and associated risks had been communicated. Overall, participants expressed a desire for transparent communication about what was known and not known rather than no communication when the accuracy of the results or the probability of an adverse outcome could not be assured. Health systems need to consider the changing landscape of access to contraception and safe abortion and the preferences of WRA and their partners for learning about the risks of emerging pathogens suspected to affect fetal or infant development before the next epidemic of an emerging infectious disease.

## Supporting information

**S1 Table. Site-level overview of dates and procedures for participant recruitment, interviews, and psychological support during and following the interview.**
(DOCX)

**S2 Table. Supporting quotes in original Spanish or Portuguese and English translation.**
(DOCX)

**S3 Table. Codebook.**
(DOCX)

**S1 Text. PLOS One Inclusivity in global research questionnaire.**
(DOCX)

## Acknowledgments

Above all, we would like to thank the participants from all sites and countries for sharing their insights and time in support of this study, especially the mothers of children living with the adverse effects of fetal exposure to ZIKV. In addition, we would like to thank Atisha A. Gómez-Reyes and Vivian Torres of the University of Puerto Rico for their support with data collection and management.

## Author Contributions

**Conceptualization:** María Consuelo Miranda Montoya, Edna Acosta Pérez, Gustavo Corrêa Matta, Olivia C. Manders, Lauren Maxwell.

**Data curation:** María Consuelo Miranda Montoya, Claudia Hormiga Sánchez, Gabriela Lopes Gama, Camila Pimentel, Marcela Mercado, Angélica María Amado Niño, Luz Marina Leegstra, Lauren Maxwell.

**Formal analysis:** María Consuelo Miranda Montoya, Claudia Hormiga Sánchez, Ester Paiva Souto, Edna Acosta Pérez, Gustavo Corrêa Matta, Marcela Daza, Gabriela Lopes Gama, Camila Pimentel, Marcela Mercado, Angélica María Amado Niño, Luz Marina Leegstra, Elena Marbán Castro, Olivia C. Manders, Lauren Maxwell.

**Funding acquisition:** María Consuelo Miranda Montoya, Edna Acosta Pérez, Marcela Daza, Lauren Maxwell.

**Investigation:** María Consuelo Miranda Montoya, Claudia Hormiga Sánchez, Ester Paiva Souto, Marcela Daza, Camila Pimentel, Marcela Mercado, Lauren Maxwell.

**Methodology:** María Consuelo Miranda Montoya, Claudia Hormiga Sánchez, Lauren Maxwell.

**Project administration:** María Consuelo Miranda Montoya, Ester Paiva Souto, Edna Acosta Pérez, Gustavo Corrêa Matta, Marcela Mercado, Olivia C. Manders, Lauren Maxwell.

**Resources:** María Consuelo Miranda Montoya, Edna Acosta Pérez, Gabriela Lopes Gama, Camila Pimentel, Lauren Maxwell.

**Supervision:** María Consuelo Miranda Montoya, Edna Acosta Pérez, Marcela Daza, Camila Pimentel, Marcela Mercado, Elena Marbán Castro, Olivia C. Manders, Lauren Maxwell.

**Validation:** Ester Paiva Souto, Edna Acosta Pérez, Lauren Maxwell.

**Writing – original draft:** María Consuelo Miranda Montoya, Claudia Hormiga Sánchez, Lauren Maxwell.

**Writing – review & editing:** María Consuelo Miranda Montoya, Claudia Hormiga Sánchez, Ester Paiva Souto, Edna Acosta Pérez, Gustavo Corrêa Matta, Marcela Daza, Gabriela Lopes Gama, Camila Pimentel, Marcela Mercado, Angélica María Amado Niño, Luz Marina Leegstra, Elena Marbán Castro, Olivia C. Manders, Lauren Maxwell.

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
