## [Decision Letter · Decision Letter 0]

6 Feb 2024

PGPH-D-23-02414

“I found out about Zika virus after she was born.” Women’s experiences of risk communication during the Zika virus epidemic in Brazil, Colombia, and Puerto Rico.

Dear Maxwel,

Thank you for submitting your manuscript to PLOS Global Public Health. After careful consideration, we feel that it has merit but does not fully meet PLOS Global Public Health’s publication criteria as it currently stands. Therefore, we invite you to submit a revised version of the manuscript that addresses the points raised during the review process.

We look forward to receiving your revised manuscript.

Kind regards,

Collins Otieno Asweto, PhD

Academic Editor

Journal Requirements:

1. We would like to request copy editing of the submission.

2. Please include a complete copy of PLOS’ questionnaire on inclusivity in global research in your revised manuscript. Our policy for research in this area aims to improve transparency in the reporting of research performed outside of researchers’ own country or community. The policy applies to researchers who have travelled to a different country to conduct research, research with Indigenous populations or their lands, and research on cultural artefacts. The questionnaire can also be requested at the journal’s discretion for any other submissions, even if these conditions are not met.  Please find more information on the policy and a link to download a blank copy of the questionnaire here: https://journals.plos.org/globalpublichealth/s/best-practices-in-research-reporting. Please upload a completed version of your questionnaire as Supporting Information when you resubmit your manuscript.

3. In the online submission form, you indicated that "De-identified transcripts of the qualitative interviews can be made available upon request."

3. Uploaded as supplementary information.

Reviewers' comments:

Reviewer's Responses to Questions

**Comments to the Author**

1. Does this manuscript meet PLOS Global Public Health’s publication criteria? Is the manuscript technically sound, and do the data support the conclusions? The manuscript must describe methodologically and ethically rigorous research with conclusions that are appropriately drawn based on the data presented.

Reviewer #1: Yes

Reviewer #2: Yes

2. Has the statistical analysis been performed appropriately and rigorously?

Reviewer #1: Yes

Reviewer #2: Yes

3. Have the authors made all data underlying the findings in their manuscript fully available (please refer to the Data Availability Statement at the start of the manuscript PDF file)?

Reviewer #1: No

Reviewer #2: Yes

4. Is the manuscript presented in an intelligible fashion and written in standard English?

Reviewer #1: Yes

Reviewer #2: Yes

5. Review Comments to the Author

Reviewer #1: The authors have done a good job in this very essential topic that brings to our attention the need to ensure that during disease epidemics, pregnant women receive appropriate, respectful and contextualized risk communication. The manuscript is well written and structured. However, I have very few areas for minor corrections which I suggest to the authors to work on as follows:

Abstract

Line 51 Methods

o Line 53: I suggest to the authors to write the ZIKV in long form and the abbreviation in brackets as follows: “………..understand women’s experiences of Zika virus (ZIKV) test…….”

110 Methods

Line 112: I suggest to edit the “ZIKA” to be ZIKV

Line 113: I suggest to write the CZS in long form first followed by the abbreviation in brackets

Line 139: “social sciences and ZIKV-focusedresearch. The guides were divided into five sections:”; I suggest to put a space between ZIKV-focused and research.

178 RESULTS

Lines 181 – 182: I suggest to use only the abbreviation CZS since its long form will be written in line 113

Line 256: I suggest to REPLACE “that” with “than” so that the sentence will read as follows: “rather than the women’s health care provider, when……”

538 DISCUSSION

Line 552: “abnormal ultrasound scan. In contest, in Puerto Rico, where the epidemic wave was later than in”; I suggest to the authors to replace “contest” with “contrast”.

Line 618: “America and the Caribbean gained experience and ministries and health services developed”; I suggest to the authors to REPLACE “and” with “of” between ministries & health so that the sentence will read as follows: “America and the Caribbean gained experience and ministries of health services developed”.

Line 628: “(SUS). Simultaneously, research initiatives were promoted with the participation of the”; I am not sure if it is clear on what does the SUS stand for? Therefore, I suggest to consider writing it in long form.

Reviewer #2: The article explored differences in diagnosis and risk communication after Zika virus exposure in 3 countries, Brazil, Columbia and Puerto Rico. The findings presented showed better preparation and response in Puerto rico than the other 2 Countries. One plausible reason given was the onset of virus being late in Puerto Rico and combined with US-CDC response made risk communication and further options informed better. It was not clear if and how abortion laws in the countries played a role. Also, data collection seemed only from pregnant women. It would have enriched if in-depth interviews were made with Ministry of Health Officials and focus group and in-depth interviews with health care providers were planned. Data from family members and wider communities enrich more information on what was known and what options were presented. As data collection happened during Covid, more information on mitigating issues would have been better. Table 1 showed participant characteristics on Educational level and marital status. Information on socio-economic status and parity would have added more perspective on the choices given and made following diagnosis. It was not clear if all the interviews were remote. What were the 7 sites from 3 Countries where data were collected? The recommendations should include how to improve accuracy of diagnosis and how to offer options after positive testing. These will include measures to address cultural and religious needs including laws and potential policy changes to authorise and legalise termination of pregnancy as a choice.

6. PLOS authors have the option to publish the peer review history of their article (what does this mean?). If published, this will include your full peer review and any attached files.

**Do you want your identity to be public for this peer review?** For information about this choice, including consent withdrawal, please see our Privacy Policy.

Reviewer #1: **Yes: **Eliudi Saria Eliakimu

Reviewer #2: **Yes: **Somasundari Gopalakrishnan

---

## [Decision Letter · Decision Letter 1]

21 May 2024

“I found out about Zika virus after she was born.” Women’s experiences of risk communication during the Zika virus epidemic in Brazil, Colombia, and Puerto Rico.

PGPH-D-23-02414R1

Dear Maxwel,

We are pleased to inform you that your manuscript '“I found out about Zika virus after she was born.” Women’s experiences of risk communication during the Zika virus epidemic in Brazil, Colombia, and Puerto Rico.' has been provisionally accepted for publication in PLOS Global Public Health.

Best regards,

Collins Otieno Asweto, PhD

Academic Editor

Reviewer's Responses to Questions

**Comments to the Author**

1. If the authors have adequately addressed your comments raised in a previous round of review and you feel that this manuscript is now acceptable for publication, you may indicate that here to bypass the “Comments to the Author” section, enter your conflict of interest statement in the “Confidential to Editor” section, and submit your "Accept" recommendation.

Reviewer #1: All comments have been addressed

Reviewer #3: All comments have been addressed

2. Does this manuscript meet PLOS Global Public Health’s publication criteria? Is the manuscript technically sound, and do the data support the conclusions? The manuscript must describe methodologically and ethically rigorous research with conclusions that are appropriately drawn based on the data presented.

Reviewer #1: Yes

Reviewer #3: Yes

3. Has the statistical analysis been performed appropriately and rigorously?

Reviewer #1: Yes

Reviewer #3: Yes

4. Have the authors made all data underlying the findings in their manuscript fully available (please refer to the Data Availability Statement at the start of the manuscript PDF file)?

Reviewer #1: No

Reviewer #3: Yes

5. Is the manuscript presented in an intelligible fashion and written in standard English?

Reviewer #1: Yes

Reviewer #3: Yes

6. Review Comments to the Author

Reviewer #1: (No Response)

Reviewer #3: It would be better to review the methodology as the results show a mixed-method approach.

7. PLOS authors have the option to publish the peer review history of their article (what does this mean?). If published, this will include your full peer review and any attached files.

**Do you want your identity to be public for this peer review?** For information about this choice, including consent withdrawal, please see our Privacy Policy.

Reviewer #1: **Yes: **Eliudi Saria Eliakimu

Reviewer #3: **Yes: **Andrew Likaka
